# Amnat Charoen Healers in Thailand and Their Medicinal Plants

**DOI:** 10.3390/plants14040602

**Published:** 2025-02-17

**Authors:** Auemporn Junsongduang, Surapon Saensouk, Henrik Balslev

**Affiliations:** 1Department of Science and Technology, Faculty of Liberal Arts and Science, Roi Et Rajabhat University, Selaphum, Roi Et 45120, Thailand; 2Walai Rukhavej Botanical Research Institute, Mahasarakham University, Kantarawichai, Maha Sarakham 44150, Thailand; surapon.s@msu.ac.th; 3Diversity of Family Zingiberaceae and Vascular Plant of Its Applications Research Unit, Mahasarakham University, Maha Sarakham 44150, Thailand; 4Ecoinformatics and Biodiversity, Department of Biology, Aarhus University Build 1540, Ny Munkegade 114, 8000 Aarhus C, Denmark; henrik.balslev@bio.au.dk

**Keywords:** ethnic group, species diversity, traditional knowledge, traditional health, Phu Tai

## Abstract

Medicinal plants remain vital in the Phu Tai community in Amnat Charoen in Thailand. Traditional healers’ knowledge is largely undocumented in the literature. Our objective was to document their medicinal plant practices to preserve this valuable knowledge. Our informants were 15 Phu Tai healers. We calculated use values (UV), family importance values (FIV), and informant agreement ratios (IAR) to gauge the significance of the 211 medicinal plants used by the healers. The most important plant families were Fabaceae and Zingiberaceae (FIV = 93). Kha min (*Curcuma longa*) was the most important medicinal species (UV = 0.66). The decoction was the most common preparation method (85%). Skin/subcutaneous cellular tissue disorders had the highest informant agreement ratio (IAR = 0.73). Shrubs were the most common life form (36%) among the medicinal plants; the majority were collected from community forests (51%) and were native to Thailand (86%). The most frequently used plant part for medicine was the leaf (27%). Medicinal plants that can be purchased were Ueang mai na (*Hellenia speciosa*), Thep tharo (*Cinnamomum parthenoxylon*), and Som khon (*Talinum paniculatum*). Interestingly, monks served as traditional healers. The healer’s age and education were not correlated with the number of medicinal plants they knew. The Amnat Charoen healers possess a rich traditional knowledge of medicinal plants. The information reported here is invaluable for further research in the field of cross-cultural ethnobotany and ethnopharmacology.

## 1. Introduction

The use of herbal medicine, rooted in the belief in plants’ healing properties, has been a longstanding practice globally [1,2,3,4]. Cultures worldwide have crafted their herbal systems, drawing wisdom and traditional healing methods from local plant knowledge [5,6,7]. Generations of indigenous healers have amassed extensive knowledge about medicinal plants through observation and experimentation [8,9,10]. The resources of medicinal plants from diverse ecosystems offer unique bioactive compounds that can benefit people’s health [11,12]. The diversity of medicinal plants across cultures enriches global herbal medicine [13,14]. Traditional healers usually possess significant local knowledge that blends medical knowledge with their cultural heritage [15]. When they inherit ancestral healing practices, they rely on natural remedies and rituals to address illnesses, often serving as primary healthcare providers where modern services are scarce [6,16]. Their important historical eras and societies are evident in China [17,18], India [19,20,21], Ethiopia [22], Iraq [23], and Thailand [24,25,26]. However, the invaluable knowledge of traditional medicine remains underappreciated in research and development [15,27]. Traditional knowledge is severely threatened by economic globalization, which often prioritizes commercial interests over cultural heritage [27,28,29,30]. Moreover, there is a risk of this wisdom fading away as older healers pass away without having produced adequate documentation of their knowledge [31,32]. Many traditional knowledge systems have been lost because they were transmitted only orally, which is vulnerable to breakdowns in intergenerational communication [33,34]. Poor relationships between older and younger generations can accelerate this loss [27,30,35]. In Thailand, traditional medicine showcases a diversity that reflects the country’s varied ecosystems and cultural practices [31,36,37,38,39]. Each region has developed unique healing methods, contributing to a rich medical knowledge [40,41,42,43]. Traditional healers in Thailand, known as “Mor Ya Puen Ban”, play a pivotal role in the healthcare landscape, particularly in rural and remote regions where access to modern medical facilities is limited [37]. The traditional culture of the healers is deeply rooted in rural communities in Thailand; they are revered for their extensive knowledge of traditional healing practices passed down through generations [25,44]. Their expertise encompasses a profound understanding of local environments and medicinal plants, which they utilize to provide comprehensive healthcare services [24]. The knowledge possessed by traditional healers is not only extensive but also highly localized, adapted to the unique environmental and cultural contexts of their regions [24,31,39]. The Phu Tai people, who are considered to be indigenous to northeastern Thailand, constitute a significant portion of the local population and possess unique traditional practices regarding medicinal plant use [45]. They coexist alongside other ethnic groups, primarily of Thai and Lao descent, in a region that is characterized by linguistic diversity, with Thai, Lao, and various ethnic languages being spoken. The majority of northeastern Thai people speak a dialect of Lao influenced by Thai, commonly referred to as Isan [45,46]. Here, we conducted a study documenting the traditional knowledge of medicinal plants of fifteen traditional healers belonging to the Phu Tai ethnic group in Amnat Charoen province.

The primary objective of this study was to identify and document the medicinal plants and associated ethnobotanical knowledge of Phu Tai traditional healers. We focused on documenting the traditional knowledge related to the utilization of medicinal plants in the treatment of a variety of ailments. The ethnomedicinal information presented here was obtained from 15 traditional Phu Tai healers in Amnat Charoen province. We hope that it will benefit those who are interested in traditional medicine and the further associated aspects of medicinal plants. In this context, and, departing, from the Phu Tai traditional healers in Amnat Charoen province, we aim to answer the following specific questions: (1) How many medicinal plants do they know and use to treat patients in their community?; (2) What is the traditional knowledge used by Phu Tai healers in Amnat Charoen province for treating their patients?; (3) Which species and plant families were most used?; (4) How many ailments were treated with traditional medicinal plants?; (5) Which parts of medicinal plants are commonly used for medical purposes?; (6) Where did the healers gather the medicinal plants?; and (7) What were the habits and statuses of medicinal plants?

## 2. Results

### 2.1. The Traditional Healers 

We documented 211 plant species in 185 genera and 70 plant families used by 15 Phu Tai healers of Chanuman district in the Amnat Charoen province of Thailand. The healers lived in four remote villages and depended on plant resources for providing primary healthcare. They had vast knowledge of the use of plants for curing various human ailments. The healers’ ages were moderately correlated with the number of medicinal plants species they knew and used (r = 0.39). Noteworthy gender disparities were observed in the use of medicinal plant species among the healers, with males exhibiting statistically significant differences compared to females (*p* ≤ 0.05); they all engaged in agriculture. Of particular interest, the highest number of known medicinal plants was reported by a 77-year-old male healer, recognized as the foremost healer in Amnat Charoen province for his profound understanding of medicinal plant applications (Table 1).

### 2.2. Family Importance Value (FIV)

Fabaceae had the highest number of species, followed by Zingiberaceae, both with FIV = 93 each, and Rutaceae with FIV = 80. Poaceae, Euphorbiaceae, and Acanthaceae each had FIV = 66 each. Additionally, Solanaceae had FIV = 46 (Appendix A).

### 2.3. Plant Parts Used

All parts of the medicinal plants were used by the healers; however, leaves, wood, and the whole plant or stem were used more often than other parts (Table 2).

### 2.4. Life Forms and Status of the Medicinal Plants

The medicinal plants used by the healers represented nine different life forms. The best-represented life form was shrubs (including undershrub, shrubby trees, exotic shrubs, and scandent shrubs). Following that were herbs (including exotic herbs, creeping herbs, and herbaceous climbers), and trees (including exotic trees). Climbers (including exotic climbers) were also prominent. The medicinal plants also included a tree palm and a climbing palm. Most of the medicinal plants used were native to Thailand, accounting for 181 species (85%), while 15% of the species of medicinal plants were exotic (Table 3).

### 2.5. Preparation and Application of Medicinal Plants Among the Healers

The healers prepared their plant medicine in eight different ways; the decoction was the overwhelmingly dominant preparation method. The most prevalent method was decoction, followed by consuming the fresh plants. Squeezing and grinding were used before applying the plants directly to the skin or drinking the extracts. A few species were prepared by soaking the plants in water, making a bolus, or by fermenting or grilling (Table 4). Less common applications included soaking the plants in the water, which were then used for bathing the patient. Even rarer application methods, mentioned for less than one percent of the species used, were inhalation for symptoms of giddiness, drips, and chewing the plants for mouth ulcers (Table 4).

### 2.6. Use Value Index (UV) Symptoms and Ailments Treated by the Phu Tai Traditional Healers

The two most significant and widely used medicinal plant species were Kha min (*Curcuma longa*; Zingiberaceae), with a use value (UV) of 0.66 and Pha ya plong thong (*Clinacanthus nutans*; Acanthaceae), with a use value (UV) of 0.53. The details of each medicinal plant are listed; UV values are listed separately (Appendix A).

### 2.7. Informant Agreement Ratio (IAR) Among the Healers

The collective knowledge of traditional healers allowed them to address a total of 46 different ailments and symptoms (Appendix A); these were divided among seventeen disorders as classified in [47]. The disorders treated with the highest degree of consensus were skin/subcutaneous cellular tissue disorders with 0.73 of IAR (Table 5, Appendix A).

### 2.8. Habitats of Medicinal Plants

The Phu Tai healers collected most of their medicinal plants in the community forests surrounding their villages (191 spp., 91%), followed by home gardens (15 spp., 7%), and a national park (Phu Sa Dong Bua, 13 spp., 6%), or they bought them from other places (3 spp., 1%).

### 2.9. Threatened Species Among the Medicinal Plants

The threatened plants list is based on IUCN, version 1994 for endemic and rare species, and version 2001 for vulnerable and endangered species. This study found the following four species with some category of threat: *Clinacanthus nutans* (rare/endemic); *Holarrhena curtisii* (endangered/vulnerable); *Lysiphyllum strychnifolium* (endangered/near threatened); and *Aquilaria crassna* (rare/critically endangered).

## 3. Discussion

### 3.1. The Traditional Healers and Their Medicinal Knowledge

The observed disparity in medicinal plant knowledge between the older healer (77 years) and the younger healer (69 years) was moderately correlated with the number of medicinal plant species they knew and used; these can be attributed to several socio-cultural and environmental factors. First, generational knowledge transfer plays a significant role in traditional healing practices. The older healer likely grew up in a time when reliance on medicinal plants was more prevalent due to limited access to modern healthcare. This context may have necessitated a more extensive and experiential understanding of local biodiversity, which was integral to community health. Conversely, the younger healer’s reduced repertoire of plant knowledge may reflect a generational shift influenced by the increased availability of modern medicine, leading to a diminished reliance on traditional remedies [30,32]. Second, socio-economic changes could contribute to the knowledge gap. Urbanization, migration, and the younger generation’s pursuit of formal education or alternative livelihoods often disrupt the intergenerational transmission of traditional ecological knowledge. Studies in similar contexts have demonstrated that elder healers act as critical knowledge repositories; however, their ability to transfer this knowledge depends on sustained interactions with apprentices or younger community members [31]. Third, traditional healers in Thailand can be classified into six distinct types, as follows: spirit healers; midwives; herbalists; ritual specialists; traditional massage; and spiritual healers. Many traditional healers are skilled in multiple practices, each with unique approaches and expertise in treating illnesses. Their knowledge and methods are shaped by ancestral wisdom passed down through generations and self-directed learning. Among these groups, herbalists stand out as the most frequent users of medicinal plants. Most of this knowledge has been passed down through generations. The healing process primarily relies on local resources, beliefs, and rituals within the community. Finally, the loss of biodiversity due to environmental degradation in rural areas might also limit the younger healer’s opportunities to practice and expand their medicinal plant knowledge. Habitat destruction reduces the availability of certain plants, thereby narrowing the scope of traditional practices [28,31].

There were fewer female than male practitioners, which may be explained by social structure and different cultures [48]. Especially in Southeast Asia, women are often limited in their roles as healers due to cultural beliefs and male-dominated healing traditions [49]. A lower female sex ratio is also found in other parts of Thailand [24,36,43] and other regions of the world, such as Latin America [50] and America, where gender roles often restrict women’s involvement in traditional medicine [51,52]. It also appears to be a general trend in the traditional Thai culture that most healers are male and elderly [31,53]. Traditionally, men have been the dominant gender and they were often seen as a symbol of strength, assuming leadership roles in both family and society [52]. This perception of men as leaders and protectors has been acknowledged by Thai society [54,55]. Gender’s influence on traditional medicinal knowledge varies widely across cultures and regions; therefore, drawing a universal conclusion about which gender generally holds more knowledge is challenging. However, research reveals some common trends in gendered knowledge distribution; in many communities, the division of labor and social roles heavily influence traditional medicinal knowledge. In Bolivian Amazon communities, men and women often have specialized knowledge relevant to their daily roles, which can create a complementary, rather than hierarchical, distribution of knowledge. For example, men may have more extensive knowledge of plants used for treating injuries or ailments associated with outdoor or physical activities such as wounds or muscular pain. women often hold specialized knowledge of plants related to childbearing, family health, and minor ailments, reflecting their roles in family care. These are examples of role-based knowledge differentiation [56]. In Afro-Brazilian communities, men have greater access to wild plants and sacred groves, while women focus on plants cultivated near the home for medicinal and daily use [57]. An analysis of plant knowledge gender differentials in northwestern Mapuche communities highlighted how men are often seen as formal healers, while women retain significant medicinal knowledge for family and reproductive health [58]. A study among Amazon communities found that gender differences in medicinal plant knowledge can vary within and between communities, indicating that gender may influence knowledge. However, this influence is not universally significant [59]. In overall conclusion, medicinal knowledge between genders is complementary, with men and women each holding unique knowledge vital for the community’s health [60]. However, the influences of gender-based disparities are minimal and not always markedly different [56,57,58,59,60,61]. Additionally, we have documented the presence of three monks who fulfill the role of traditional healers within their respective communities. Monks hold a vital position in rural communities, particularly in the northeastern regions of Thailand [62]. Historically, monks have had a crucial role in the daily lives of people in northeastern Thailand. Monks have an impact on social cohesion, cultural preservation, and spiritual well-being [63,64]. Consequently, this discovery underscores the enduring significance of monks in remote rural societies in northeastern Thailand, demonstrating their continued role and relevance to this day.

### 3.2. Diversity of Medicinal Plants

Among the various families of medicinal plants, Fabaceae and Zingiberaceae were the most used, as reflected by the high number of species and the FIV index. These results are in general agreement with previous investigations in Thailand showing that, among ethnic groups in Thailand, the most prominent family for medicinal plants was Fabaceae [65]. The same was true in northeastern Thailand, among, Phu Tai healers in Roi Et province [31], Nakhonpanum province [38], and Kalasin province [66]. In the south of Thailand, the most commonly used plant families were again Fabaceae and Zingiberaceae [43]. Similar results are reported in studies in other parts of the world, such as China [67], and Namibia [68], where Fabaceae also had the highest number and percentage of medicinal plant species. Fabaceae’s global dominance, as seen in Thailand, Namibia, and parts of India, is attributed to its ecological versatility, nitrogen-fixing ability (allowing it to grow in poor soils), and rich secondary metabolites. Its species often exhibit wide pharmacological activity, making it a universally relied-upon family for traditional medicine. Similarly, Zingiberaceae’s aromatic rhizomes have proven effective across tropical regions for digestive and anti-inflammatory uses, reflecting their adaptation to these climates and associated diets. However, in Iraq, Lamiaceae was the family with the most medicinal plant species [23]. It is likely related to its adaptability to the arid and semi-arid climates that characterize much of the region. Lamiaceae species, such as mint and thyme, thrive in these conditions and are widely recognized for their essential oils, which have antimicrobial and anti-inflammatory properties [23]. In India, the Euphorbiaceae family, prominent in Kerala [21]. and Tamil Nadu [69], is ecologically abundant in tropical and subtropical regions. The family includes species like Phyllanthus and Euphorbia, which are known for their pharmacological properties, including anti-diabetic, hepatoprotective, and anti-inflammatory effects [70]. The diverse uses of Euphorbiaceae in India also reflect its cultural integration into Ayurvedic medicine, where its species are used for treating chronic diseases. In Morocco, the prominence of Asparagaceae, specifically for dermatological conditions [71], is likely tied to the region’s Mediterranean climate. Species, such as *Asparagus officinalis* and *Dracaena*, thrive in this environment and are rich in saponins and antioxidants, which are effective in skin-related therapies [72]. The prominence of certain plant families in traditional medicinal systems is influenced by a combination of climatic conditions, ecological availability, cultural practices, and the specific therapeutic needs of local populations. This demonstrates how environmental conditions and prevalent health issues shape the medicinal plant preferences of local populations.

### 3.3. Parts of Plants Used as Medicine

In this study, leaves were the most commonly utilized part of medicinal plants, aligning with findings among the Phu Tai ethnic group in Nakhon Phanom province, Thailand [38]. However, this result contrasts with the findings of Junsongduang et al. [31], where stems were predominantly used by Phu Tai traditional healers in Roi Et province. Regional differences are further evident in southern Thailand, where underground parts, such as roots, rhizomes, tubers, and corms, are the primary choice for medicinal purposes [24]. A similar preference for leaves has been observed in India [21,69], highlighting the widespread reliance on this plant part for traditional medicine.

The preference for leaves may be attributed to their ability to regenerate, allowing for sustainable harvesting that does not compromise plant survival. In contrast, harvesting roots, stems, or bark often poses greater risks to plant health and survival, particularly with repeated extraction [73]. Such sustainable practices reflect ecological awareness and traditional knowledge systems within communities. Interestingly, our findings differ from studies in China, where the whole plant is frequently utilized, particularly among herbaceous medicinal plants [67]. The use of entire plants may stem from their small size and the practicality of using them holistically. Additionally, the multiple uses of individual plants in various medicinal applications highlight the depth of traditional knowledge and the versatility of local flora. These variations across regions underscore the influence of ecological, cultural, and practical considerations in shaping the use of plant parts for medicinal purposes [74].

### 3.4. Life Forms of Medicinal Plants

A total of nine different life forms of medicinal plants were identified among the healers. The most commonly used life form was shrubs, which is consistent with a study conducted in southern Thailand [24]. However, this finding contrasts with another study in northeastern Thailand, specifically in Nakhon Phanom and Roi Et provinces, where trees were the most prevalent life form used for medicinal purposes [31]. The advantage of trees lies in their multiple usable parts, which can be harvested repeatedly, allowing for sustainable collection. In contrast, herbs, which are used in their entirety, are typically consumed once and may be at risk of overharvesting, potentially leading to extinction [31]. Following shrubs, herbs, including both native and exotic species and herbaceous climbers, were commonly utilized. Additionally, two species of palms, including a climbing palm, were recognized for their medicinal properties. Unique to this study, one species each from the life forms of bamboo, exotic aquatic herb, and epiphytic fern were also documented as having medicinal uses. For comparative purposes, studies in Guangxi Fangcheng, China, and Tamil Nadu, India, found that herbs were the predominant life form used in traditional medicine [67,69], highlighting regional differences in plant selection based on ecological and cultural factors.

### 3.5. Preparation and Application of Medicinal Plants

Altogether, the Phu Tai healers used a total of eight different methods for preparing and applying the medicinal plants. The most common preparation method was decoction for internal use, which is similar to methods used elsewhere in Thailand [65]. Decoctions were used to treat a variety of conditions, including digestive diseases [43], metabolic diseases [31], and blood system disorders [36]. They are also often used in women’s healthcare [75]. Following decoction, the next most important way of preparing the medicine was to consume the fresh plants. Grinding and applying the plants directly to the skin, as well as squeezing and drinking the extracts, were also common practices among the healers. Soaking the plants in water and drinking them or using the extraction for a bath was also found among Phu Tai healers in Roi Et province the same ethnic group [31]. Making a bolus, or fermenting and grilling, were documented in this study but were not observed in Roi et [31]. The most common application method was drinking. This was also in line with the most common preparation being a decoction. Inhaling, drops, and chewing were rare methods that were mentioned for either two, or only one single medicinal plant, and accounted for less than one percent of the total species used. External use was rare; in these cases, parts of the medicinal plants were usually compressed, smeared, and then applied to the affected area, a common way to treat skin diseases, muscle pain, swellings, and wounds. Most of the remedies used a single plant part and more than one method of preparation. However, many of the remedies consisted of different parts of the same plant species used to treat more health conditions. For example, the tree Daeng (*Xylia xylocarpa:* Fabaceae) was used to treat pain; the bark was used to treat fever; the root was used to treat diarrhea; and the leaf was boiled for a bath to treat flatulence. The whole plant of Ma khet (*Catunaregam tomentosa;* Rubiaceae). was decocted to treat liver cancer, fever, and used as a diuretic. The stem of the climber *Connarus semidecandrus* was used to treat fever; its roots were used to treat diarrhea; and its leaves were used to treat gastritis. The rhizome of Kha min (*Curcuma longa;* Zingiberaceae) was decocted for treating flatulence, allergy, and uterus conditions. The root of Mo noi (*Cyanthillium cinereum;* Asteraceae) was ground with water and drunk for fever and tonic; decoctions of its flowers were used to treat gastritis.

### 3.6. Use Value (UV), Symptoms, and Ailments Treated by Phu Tai Traditional Healers

The use value serves as a quantitative measure that indicates the relative significance of species [76]. The two most significant and widely used medicinal plant species among Phu Tai healers were Kha min (*Curcuma longa*; Zingiberaceae) and Pha ya plong thong (*Clinacanthus nutans*; Acanthaceae). This finding agrees with many reports on the potential of the two medicinal plants in medicinal use. Turmeric, known scientifically as *Curcuma longa*, has been a key component of Ayurvedic medicine for centuries and is valued for both its medicinal and dietary uses. The main active compound in turmeric, curcumin, has been extensively researched and proven to be non-toxic to humans. It is recognized for its antioxidant, anti-inflammatory, antiviral, and antifungal properties [77]. *Curcuma longa* is a wild species of the ginger family which has its highest diversity of species in southeast Asia [78]. In addition to its medicinal uses, it is commonly used as a spice; it is the most important spice in many dishes in southeast Asia including in Thailand [79,80,81,82]. In Asian traditional medicine, this species is well known; it is frequently used by traditional healers, for example, in India [71], for curing diarrhea, stomach aches, skin care, facial massage, arthritis pain, fat-burning, spice, antiviral, and anticancer agents [23]. In Thailand it is used for pain relief in the northeast [83], while in the south of Thailan, it is used to treat peptic ulcers, intestinal infections, and skin diseases [24]. Historically, it has been employed to address a range of health issues, including digestive, respiratory, and circulatory ailments, and various skin conditions. Recognizing its significance in global health practices, the establishment of an international standard for *Curcuma longa* has become imperative to ensure its effectiveness, safety, and manageability in worldwide commerce and trade [84]. However, in different regions in Thailand, other species had the highest UV. For example, in northeastern Thailand in Roi Et province, Mak mor (*Rothmannia wittii*; Rubiaceae) was the most used species [31]. In Songkhla province, in southern Thailand, Khing (*Zingiber officinale;* Zingiberaceae) had the highest use value while in the Krabi province, it was Dee pree (*Piper retrofractum*; Piperaceae) [43]. In the north, *Leonotis nepetifolia;* Lamiaceae and *Schefflera* cf. *bengalensis*; Araliaceae, were the medicinal species most used for treating postpartum women among the traditional healers in Nan province [85]. In other countries, different species were the most used. In India [21], Maroom *(Moringa oleifera*; Fabaceae) is supreme in the lush valleys of Kerala. Khing (*Zingiber officinale*; Zingiberaceae) had the highest UV in Iraq [23]. In Roi Et, a single medicinal plant species is popular for use in the treatment of more than one disease. For Ma khet (*Catunaregam tomentosa*; Rubiaceae), the whole plant is decocted for curing liver cancer, fever, and diuretics. A decoction of the stem or root or leaf of Thop thaep khruea (*Connarus semidecandrus*; Connaraceae) is used to treat fever, diarrhea, and gastritis. The leaves and bark of Plao yai (*Croton persimilis*; Euphorbiaceae) are used to make a compress for treating muscle pain or the fruit is eaten fresh to treat diabetes. A decoction of the root, leaf, and stem of Katang bai (*Leea indica*; Vitaceae) is used to treat gallstones, other kidney diseases, and hemorrhoids. A decoction of the root, or the whole plant of Oi dum (*Saccharum x sinense*; Poaceae), is used to treat gall stones, hemorrhoids, and gastritis. Accordingly, medicinal plants with high UV should be screened for biological and antibacterial activity [86]. Additionally, the isolated compounds from the most frequently cited plants could serve as valuable starting points for the development of new drugs in the future [87].

### 3.7. Informant Agreement Ratio (IAR)

The informant agreement ratio was used to evaluate the homogeneity or consensus of knowledge among the information given. In our data, skin/subcutaneous cellular tissue disorders and muscular skeletal system disorders had the highest IAR value among all treated disorders. The high IAR value for the treatment of skin diseases by Phu Tai traditional healers may be due to the local climate and livelihoods in northeast Thailand. Local people live in rural areas; their daily livelihood is primarily manual labor, which is often very heavy. This, coupled with the long-term hot climate, means that muscular and skeleton system diseases, and skin diseases, are common [39]. The frequent occurrence of skin diseases is also found in other hot and humid climate zones such as in China [67], India [21], Turkey [88], and Namibia [68]. However, the study among Phu Tai traditional healers in Roi Et province found that the highest IAR (reported as informant consensus factor ICF) was jaundice [31]. Among the Akha communities in China, blood system disorders had the highest IAR values, while the use categories with the highest consensus of Akha communities in Thailand were pregnancy/birth/puerperium disorders [36]. At a more general level, it appears that villages inhabited by the same ethnic group, and even living within the same floristic regions, may have unique ethnobotanical knowledge [65]. Ethnic groups that migrated to Thailand centuries ago now use plants differently for medicine. This change in traditional knowledge may be attributed to adaptation, and differences in occupation, geography, and daily routines, which have led to the development of new knowledge within each community. Furthermore, the high IAR values (often reported as the informant consensus factor, ICF) in each category depend on the objective of the study and on its focus. For example, Maneenoon et al. [43] studied the knowledge of traditional healers on the use of medicinal plants to treat menstrual disorders in Krabi and Songkhla provinces in southern Thailand. They reported that the highest IAR value was for menstrual disorders. The study on medicinal plants of the Mien (Yao) in northern Thailand and their potential value in the primary healthcare of postpartum women found that the use category with the most use was that of birth-related conditions [85]. The ethnomedicinal plants used to treat digestive system disorders by the Karen of northern Thailand [42] had the highest ICF values for carminative disorders, stomachaches, geographic tongue, constipation, appetite stimulants, and food poisoning. The consensus knowledge of medicinal plants among the informants in each area depended on the scope and objective in each study. The high ICF values observed across all categories suggest that medicinal plant species for disease treatment are relatively concentrated and exhibit a high level of consistency [67]. High ICF values suggest that the species traditionally used to treat each ailment would need additional research into their phytochemical and pharmacological or bioactive compounds [68].

### 3.8. Habitat, Status and Threatened Medicinal Species

The Phu Tai healers residing in Amnat Charoen primarily sourced their medicinal plants from the community forests surrounding their villages, followed by home gardens and the nearby national park. These findings were the same as findings from studies in Roi Et [31] and Phattalung provinces [24], where the medicinal plants utilized by traditional healers continued to be harvested from wild sources. To alleviate the pressure on wild plant populations, the preservation of medicinal plants should be undertaken in community and kitchen gardens [86]. This would ensure a sustainable supply of safe, effective, and affordable medicinal plants among their communities. In cases where cultivated plants were used, their original habitats were typically distant from the traditional healers’ villages, prompting some plants to be relocated and grown in fields or home gardens for easier access [24,54]. Notably, a study by Tanjittman et al. [42] revealed that local environmental conditions, the availability of medicinal plants, and the proximity of ethnic villages to the nearest urban centers exerted a substantial influence on the depth of traditional medicinal knowledge within each village. We found that traditional healers are increasingly concerned about dwindling access to medicinal plants. Some species have become scarce or even locally extinct due to excessive harvesting and ongoing deforestation [24]. This trend is also found in China, where over 93% of medicinal plants were sourced from the wild [67]. Despite their integral role in traditional healing, these plants face habitat loss due to their many uses. Additionally, there was limited reliance on external sources, with only three species (*Cinnamomum parthenoxylon*, *Hellenia speciosa*, and *Talinum paniculatum*) being obtained from other locations. Phu Tai healers resort to purchasing these plants from distant areas because they are disappearing from local forests and surroundings.

Among the medicinal plants used by Phu Tai healers, several exotic species were identified, representing 15% of the total species. These exotic plants, including shrubs, herbs, climbers, and other life forms, are predominantly grown in home gardens or are sourced from community forests and, in rare cases, national parks. Examples include *Andrographis paniculata* (native to India, Nepal, and Sri Lanka), *Aloe vera* (originating from Northern Oman and Africa), and *Catharanthus roseus* (from Madagascar), which are cultivated in home gardens for their accessibility and ease of use. Exotic shrubs, such as *Barleria prionitis* (native to Tropical Africa, Madagascar, and Asia and *Punica granatum* (originating from Türkiye to Northern Pakistan), are grown primarily in home gardens. They are valued for treating digestive system disorders, muscular pain, and other ailments. Herbs like *Artemisia vulgaris* (native to Eurasia, Indochina, and North Africa) and *Chromolaena odorata* (native to Tropical and Subtropical America) are also common, with their parts harvested for treating sensory system disorders, injuries, and women’s health conditions. The relocation of these species from their original habitats to home gardens is often due to their dwindling availability in the wild, resulting from overharvesting and habitat destruction. In some cases, exotic plants, such as *Cestrum nocturnum* (native to Mexico and Venezuela), are purchased from distant areas to meet local medicinal needs, as they have become scarce in community forests. The introduction of these plants reflects both their perceived medicinal value and the adaptability of traditional healers in integrating foreign species into their practices.

Furthermore, four of the medicinal plants encountered in this study turned out to be subject to some threats. Threatened, endemic, rare, vulnerable, and endangered species were found in this study [89]. Steps taken by various government departments and NGOs in this direction in recent years would definitely strengthen traditional healthcare systems.

## 4. Materials and Methods

### 4.1. Study Area

Northeastern Thailand, commonly known as Isan, is home to over 20 million people, who make up approximately 30% of Thailand’s total population [90]. The predominant language spoken in this region is Isan, an Austro–Thai language closely related to Lao that is estimated to be spoken by 15–23 million people [46,90]. Northeastern Thailand is divided into 19 provinces. It is geographically limited by the Mekong River to the north and east, the Dong Rek mountain range to the south, and the Petchabun mountain range to the west. These geographical boundaries serve to separate northeastern Thailand from neighboring countries, such as Lao P.D.R. and Cambodia, as well as the northern and central parts of Thailand (Figure 1). Despite these distinct borders, the population of northeastern Thailand embodies a richness of ethnicities and cultures that is influenced by their neighbors. The northeastern Thai population is composed of over 15 different ethnic groups, with Lao and Khmer being the largest among them [46,91]. This diverse ethnic composition contributes to the unique culture of the region, enriching its traditions, practices, and ways of life. The people of northeastern Thailand face economic challenges, as they have the lowest income in the country, earning an average of 20.271 Baht per month in 2017 [90]. While agriculture remains the primary occupation, some individuals seek additional sources of income by working outside the region. Agriculture plays a vital role in the economy of northeastern Thailand, constituting the largest sector and contributing approximately 22% to the gross regional product [90]. Rice cultivation dominates the agricultural landscape, covering about 60% of the cultivated land. Despite these economic factors, the population of northeastern Thailand maintains a distinct ethnic and cultural identity, influenced by the neighboring regions.

Amnat Charoen province covers 3290 km^2^ (Figure 1) and includes both rural and urban areas. It is home to a diverse population consisting of various ethnic groups, with the majority being of Thai and Lao descent. The total population in 2019 was 378,438. The population density was 115/km^2^. The province lies on the southern bank of the Mekong River, which forms the natural boundary between Thailand and Laos. This study was undertaken in the Chanuman district, 85 km from Mueang Amnat Charoen province and 680 km from Bangkok. Chanuman district, in Amnat Charoen, Thailand, is approximately centered at a latitude of 16.186614° N and a longitude of 105.0067969° E. Given the district’s area, it spans approximately from 16.1458061° N to 16.234° N in latitude and from 104.8921668° E to 105.0067969° E in longitude. The largest important water resource is the Mekong River. Chanuman is the district where Tai Lao and Phu Tai ethnic groups have settled. However, most of the population is of the Phu Tai ethnic origin. We gathered traditional knowledge concerning medicinal plants from 15 traditional healers. These healers reside in four different villages within the Chanuman district, which is home to the largest population of Phu Tai. The traditional healing practices in this region continue to thrive and hold significant importance, even with the presence of a subdistrict health-promoting hospital.

### 4.2. Informants

In the Chanuman district of Amnat Charoen province (Figure 1), 15 traditional PhuTai healers who actively practiced herbalism were identified to us by the village chairmen in the villages Kam Doi (3 healers), Sai Na Dong (3 healers), Bung Kaew (7 healers), and Na Pooe (2 healers) (Table 1). These individuals were aged between 48 to 80 years old and were primarily involved in agriculture; three of them were also ordained monks. All 15 healers adhered to the Buddhist faith and had received only a primary education (Table 1). It is worth noting that they continue to carry out healing practices parallel to their primary occupations.

### 4.3. Data Collection

Data and plant specimens were collected from September 2021 to August 2022. Traditional healers were purposefully selected for inclusion in the study using both purposive sampling and snowball sampling techniques [92]. Initially, purposive sampling was employed, where local villagers and community members were consulted to identify recognized and well-regarded traditional healers who had expertise in using medicinal plants. These individuals were selected based on their reputation, experience, and specialized knowledge in traditional healing. Following the purposive selection, the snowball sampling technique was used to expand the pool of participants. Once initial healers were identified, they were asked to refer other healers in the community, thus enabling the study to build a network of participants. This technique ensured that healers with different levels of expertise and from diverse backgrounds could be included. Visits were made to the healers’ homes, where informal meetings were held in the healers’ preferred languages such as Lao or Phu Tai. While other healers were present in the villages, the inclusion criteria ensured a diverse representation of traditional healing knowledge. Semi-structured interviews were conducted to gather both qualitative and quantitative data on medicinal plants [44]. The interviews, which delved into the healers’ knowledge, training, treatment techniques, and preparation methods, were supplemented by direct observations. Additionally, field walks were undertaken to search for medicinal plants and assess their availability in various habitats within and around the villages [93]. Information on local plant names, parts used for treatments, preparation methods, administration routes, and plant habitats were recorded. Voucher specimens were collected for botanical identification. Initial identifications of medicinal species were made in the field based on their common names, referencing Tem Smitinand’s Thai Plant Names [94]. These specimens were then stored at the Department of Science and Technology, Roi Et Rajabhat University, for the documentation of their botanical identity. Further identification and taxonomic confirmation were carried out at the herbarium of the Queen Sirikit Botanical Garden (QSBG) using the Flora of Thailand treatments and by comparing them with existing collections, facilitated by the plant taxonomist in QSBG (WT). The categorization of ailments and symptoms followed Cook [47]. The original data or native ranks of exotic medicinal plants in this study were categorized according to POWO.

### 4.4. Data Analysis

In this study, we define a “Use report” as the specific mention of a plant being used for a particular purpose by an individual informant in a specific location at a given time. The use value serves as a quantitative measure indicating the relative significance of a species. The Use Value index (UV) [76] is expressed as follows:UV = Ui/N (1)
where Ui represents the number of use reports attributed to each informant for a given species and N denotes the total number of informants. UVs tend to be low (approaching 0) when there are few use reports associated with the use of a species. Conversely, UVs are high (approaching 1) when numerous use reports exist for a species, indicating widespread knowledge among informants.

The informant agreement ratio serves as an index for evaluating the homogeneity of knowledge [95], as follows:IAR = Nur − Nt/Nur − 1 (2)
where Nur represents the number of use reports for a specific use category, and Nt denotes the number of taxa used for that particular use category by all informants. Low IAR values (nearing 0) suggest random plant usage or a minimal exchange of information regarding their use among informants. Conversely, high IAR values (nearing 1) indicate well-defined selection criteria within the community and/or extensive information exchange among the informants.

Family importance value (FIV) gives local importance to the families of plant species. It was calculated as the percentage of informants mentioning a specific family [96], as follows:FIV = FC (family)/N × 100 (3)
where FC is the number of informants mentioned in the family and N is the total number of informants who participated in the study.

## 5. Conclusions

Knowledge of medicinal plants for treating a large number of health conditions remains an important part of the traditional culture in northeastern Thailand, where traditional healers use 211 different plant species to treat their patients. During our interviews, our informants expressed concern about the overexploitation of medicinal plants, which has led to an increasing cultivation in home gardens. Some medicinal plants were threatened, especially *Clinacanthus nutans*, *Holarrhena curtisii*, *Aquilaria crassna*, and *Lysiphyllum strychnifolium* [97]. As in other parts of Thailand and southeast Asia, certain plant families, such as Fabaceae (legumes) and Zingiberaceae (gingers), are prominent among medicinal plants; as demonstrated elsewhere, the leaves are the most commonly used part of the plant used for the making of medicine. We advocate for the further scientific exploration of specific plants, such as *Albizia myriophylla*, *Bauhinia strychnifolia*, *Eurycoma longifolia*, and *Zingiber cassumunar*, based on their high use values (UV). These medicinal plants could be novel drug candidates as potential sources for the discovery and development of new medicines, highlighting the intersection between traditional wisdom and modern pharmacological research.

## Figures and Tables

**Figure 1 plants-14-00602-f001:**
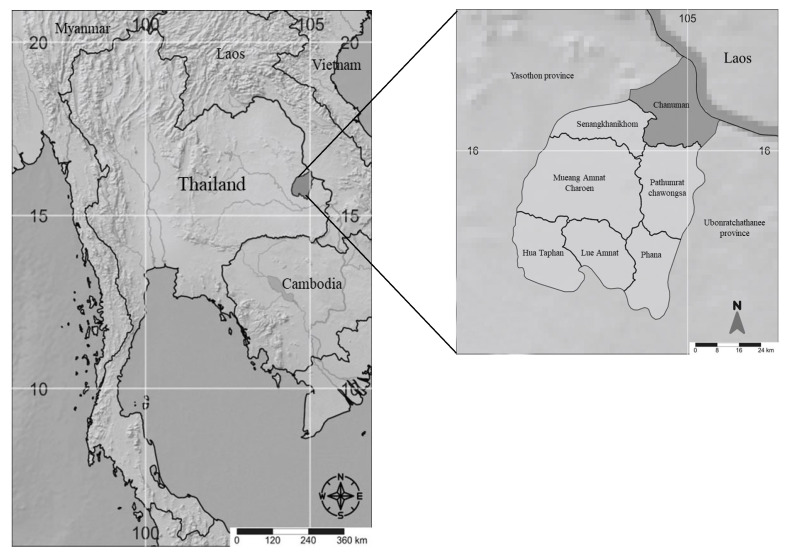
Location of Amnat Charoen province in Thailand where ethnobotanical data concerning medicinal plants were collected in Chanuman district.

**Table 1 plants-14-00602-t001:** Profile of the 15 Phu Tai healers interviewed in Chanuman district, Amnat Charoen province, and the numbers of medicinal plants known and used by each of them.

Healer No.	Gender	Age (years)	Occupation	Village	Number of Medicinal Plants Known and Used
1	Male	77	Agriculture	Kam Doei	130
2	Male	69	Agriculture	Kam Doei	21
3	Male	55	Agriculture	Kam Doei	21
4	Male	66	Agriculture	Sai Na Dong	22
5	Male	73	Agriculture	Sai Na Dong	28
6	Male	54	Agriculture	Sai Na Dong	18
7	Male	80	Agriculture	Bung Kaew	27
8	Male	60	Monk	Bung Kaew	20
9	Male	80	Monk	Bung Kaew	19
10	Male	70	Monk	Bung Kaew	19
11	Female	55	Agriculture	Bung Kaew	17
12	Female	48	Agriculture	Bung Kaew	17
13	Female	67	Agriculture	Bung Kaew	22
14	Female	64	Agriculture	Na Poe	14
15	Female	62	Agriculture	Na Poe	13
Average		63			27

**Table 2 plants-14-00602-t002:** Number of medicinal plant species of which different plant parts were used medicinally by Phu Tai healers in Amnat Charoen province.

Part Used	No. of spp.	%
Leaf	68	27
Wood	35	14
Whole	34	13
Stem	31	12
Bark	23	9
Fruit	20	8
Rhizome	19	7
Root	10	4
Flower	5	1.96
Seed	4	1.56
Shoot	2	0.78
Gum	2	0.78
Tuber	2	0.78
	255	100

**Table 3 plants-14-00602-t003:** Life forms and statuses of the medicinal plants used.

Life Form	Species (%)	Native spp.	Exotic spp.
Shrubs	75 (36)	62	13
Herbs	48 (23)	37	11
Trees	39 (19)	38	1
Climbers	38 (18)	36	2
Grasses	6 (3)	3	3
Palms	2 (1)	2	0
Aquatic herb	1 (0.5)	1	0
Bamboo	1 (0.5)	1	1
Epiphyte fern	1 (0.5)	1	0
Total	211 (100)	181	31

**Table 4 plants-14-00602-t004:** Preparation and application methods for medicinal plants used by fifteen healers in Chanuman district, Amnat Charoen, Thailand.

Preparations	No. of spp.	%	Applications	No. of spp.	%
Decoction	176	71	Drinking	198	73
Fresh	29	12	Eating	35	13
Squeezing	19	4	Smearing	15	6
Grinding	23	9	Compress	13	5
Soaking	5	2	Bath	3	1
Bolus	2	0.8	Inhale	2	0.76
Fermenting	1	0.4	Chew	1	0.38
Grilling	1	0.4	Eye and ear drops	2	0.76

**Table 5 plants-14-00602-t005:** Disorders treated and the informant agreement ratio (IAR) (homogeneity of use) of medicinal plants used by Phu Tai traditional healers.

Disorders Treated [47]	Use Reports (Ui)	No. of spp. (Nt)	IAR
Skin/Subcutaneous Cellular Tissue Disorders	39	11	0.73
Muscular Skeletal System Disorders	53	24	0.55
Nutritional Disorders	63	32	0.5
Pregnancy/Birth/Puerperium Disorders	40	22	0.46
Poisonings	10	6	0.44
Digestive System Disorders	90	54	0.40
Infections/Infestations	80	50	0.37
Sensory System Disorders	20	13	0.36
Endocrine System Disorders	13	9	0.33
Genitourinary System Disorders	16	11	0.33
Circulatory System Disorders	25	17	0.33
Respiratory System Disorders	17	12	0.31
Neoplasms	24	17	0.30
Inflammations	6	5	0.2
Injuries	10	9	0.11
Abnormalities	2	2	0.00
Nervous System Disorders	2	2	0.00

## Data Availability

Data is contained within the article or Appendix A.

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
