# Peer review of "Amnat Charoen Healers in Thailand and Their Medicinal Plants"

_plants, 2025, doi:10.3390/plants14040602_

Round 1
Reviewer 1 Report
Comments and Suggestions for Authors
1. Some information is unnecessarily repeated, e.g. the number of healers - lines 86, 120, 161, 268, 282, 454, 460, 464.
2. The content of the tables is discussed in the text. Since everything is included in the tables, the text should include what the reader does not read in the tables. As a result, the text of the paper is unnecessarily extended.
3. What is the purpose of the digression about spice plants? – lines 322-324.
4. There are two subchapters 3.6 in the text – the numbering of the subchapters should be corrected.
5. Are there no estimates from recent years? – line 438
6. Are there no later population censuses? – line 447
7. Since the district occupies some area, the geographical coordinates of only one point cannot be determined. It is better to specify between which degrees of latitude and longitude the district is located – lines 450-451.
8. "Conclusion" partially repeats "Results", e.g. lines 539-548. This should be changed.
Author Response
Comments 1. Some information is unnecessarily repeated, e.g. the number of healers - lines 86, 120, 161, 268, 282, 454, 460, 464.
Response 1: We have edited the text t avoid such repetitions. For the number of healers we have kept the number first time they are mentioned in the results and again in the methods section which would otherwise not make sense.
Comments 2. The content of the tables is discussed in the text. Since everything is included in the tables, the text should include what the reader does not read in the tables. As a result, the text of the paper is unnecessarily extended.
Response 2: We have now edited the text so all repetition of data presented in the tables is avoided.
Comments 3. What is the purpose of the digression about spice plants? – lines 322-324.
Response 3: In line 337 in the revised manuscript, we have added “In addition to it medicinal uses, ….”. We wish to explain how important and popular of Curcuma longa is and used in different ways more than medicinal purposes. This species is popular for food like used as a spice and also these plants have medicinal properties. That is why this plant had the highest UV in medicinal use also.
Comments 4. There are two subchapters 3.6 in the text – the numbering of the subchapters should be corrected.
Response 4: The numbering of subchapters has been corrected.
Comments 5. Are there no estimates from recent years? – line 438
Response 5: Line 517 edited “As of 2017, the average monthly household income in northeastern Thailand was reported at 20,271 Thai Baht.” (Ref.; Thailand Avg Monthly Household Income: Northeastern | Economic Indicators | CEIC)
Comments 6. Are there no later population censuses? – line 447
Response 6: Line 526-527 the latest Edited “The total population in 2025 was 378,438. The population density was 115/km2.” Ref. (https://en.wikipedia.org/wiki/Chanuman_district )
Comments 7. Since the district occupies some area, the geographical coordinates of only one point cannot be determined. It is better to specify between which degrees of latitude and longitude the district is located – lines 450-451.
Response 7: We edited -line 474-476. “Chanuman District in Amnat Charoen, Thailand, is approximately centered at a latitude of 16.186614° N and a longitude of 105.0067969° E. Given the district's area, it spans approximately from 16.1458061° N to 16.234° N in latitude and from 104.8921668° E to 105.0067969° E in longitude”. In addition coordinates are shown in Fig. 1, which is a map of the area.
Comments 8. "Conclusion" partially repeats "Results", e.g. lines 539-548. This should be changed.
Response 8: We have edited the conclusion so it does not repeat the results of the study.
Reviewer 2 Report
Comments and Suggestions for Authors
The manuscript is interesting and well written. I just have a few clarifying questions.
In the method, you explain in one sentence how you found the informants: "Traditional healers were purposefully selected for inclusion in the study and [89], visits were made to their homes where informal meetings were held, often in either Lao or Phu Tai languages.“
89. Bernard, H.R. Social Research Methods: Qualitative and quantitative approaches. 2nd ed. Thousand Oaks, CA: SAGE Publications, Los Angeles, 2013.
You should explain more in detail how you found these healers? Did you ask the villagers who the healer is? You should explain better why you chose this particular geographical area for your research? There is also no explanation as to whether there are other healers in these villages besides the ones you interviewed? You also lack information on the availability of academic doctors and hospitals in this region?
I have a question about the results (see table 1): You need to explain why the first healer has almost 10x more plants than the others? What caused this, was the survey method different? This extremely high number is not statistically reliable compared to what other healers said.
Author Response
Comments 1: The manuscript is interesting and well written. I just have a few clarifying questions.
In the method, you explain in one sentence how you found the informants: "Traditional healers were purposefully selected for inclusion in the study and [89], visits were made to their homes where informal meetings were held, often in either Lao or Phu Tai languages.“
- Bernard, H.R. Social Research Methods: Qualitative and quantitative approaches. 2nd ed. Thousand Oaks, CA: SAGE Publications, Los Angeles, 2013.
You should explain more in detail how you found these healers? Did you ask the villagers who the healer is? You should explain better why you chose this particular geographical area for your research? There is also no explanation as to whether there are other healers in these villages besides the ones you interviewed. You also lack information on the availability of academic doctors and hospitals in this region.
Response 1; More explanations were added. Data and plant specimens were collected from September 2021 to August 2022. Traditional healers were purposefully selected for inclusion in the study using both purposive sampling and snowball sampling techniques [91]. Initially, purposive sampling was employed, where local villagers and community members were consulted to identify recognized and well-regarded traditional healers who had expertise in using medicinal plants. These individuals were selected based on their reputation, experience, and specialized knowledge in traditional healing. Following the purposive selection, the snowball sampling technique was used to expand the pool of participants. Once initial healers were identified, they were asked to refer other healers in the community, thus allowing the study to build a network of participants. This technique ensured that healers with different levels of expertise and from diverse backgrounds could be included. Visits were made to the healers' homes, where informal meetings were held in the healers' preferred languages, such as Lao or Phu Tai. While other healers were present in the villages, the inclusion criteria ensured a diverse representation of traditional healing knowledge.
Comments 2: I have a question about the results (see table 1): You need to explain why the first healer has almost 10x more plants than the others? What caused this, was the survey method different? This extremely high number is not statistically reliable compared to what other healers said.
Response 2: More explain was added. “ The observed disparity in medicinal plant knowledge between the older healer (77 years) and the younger one (69 years) was moderately correlated with the number of medicinal plants species they knew and used and they can be attributed to several socio-cultural and environmental factors. First, generational knowledge transfer plays a significant role in traditional healing practices. The older healer likely grew up in a time when reliance on medicinal plants was more prevalent due to limited access to modern healthcare. This context may have necessitated a more extensive and experiential understanding of local biodiversity, which was integral to community health. Conversely, the younger healer’s reduced repertoire of plant knowledge may reflect a generational shift influenced by the increased availability of modern medicine, leading to a diminished reliance on traditional remedies [30,32]. Second, socio-economic changes could contribute to the knowledge gap. Urbanization, migration, and the younger generation's pursuit of formal education or alternative livelihoods often disrupt the intergenerational transmission of traditional ecological knowledge. Studies in similar contexts have demonstrated that elder healers act as critical knowledge repositories, but their ability to transfer this knowledge depends on sustained interaction with apprentices or younger community members [31]. Third, traditional healers in Thailand can be classified into six distinct types: spirit healers, midwives, herbalists, ritual specialists, traditional massage, and spiritual healers. Many traditional healers are skilled in multiple practices, each with their unique approaches and expertise in treating illnesses. Their knowledge and methods are shaped by ancestral wisdom passed down through generations and self-directed learning. Among these groups, herbalists stand out as the most frequent users of medicinal plants. Most of this knowledge has been passed down through generations. The healing process primarily relies on local resources, beliefs, and rituals within the community. Finally, the loss of biodiversity due to environmental degradation in rural areas might also limit the younger healer’s opportunities to practice and expand their medicinal plant knowledge. Habitat destruction reduces the availability of certain plants, thereby narrowing the scope of traditional practices [28,31].”
Reviewer 3 Report
Comments and Suggestions for Authors
The authors have done a detailed study of the use of medicinal plants in a traditional community in Thailand. The study is significant as it gives good details about the use of plants and what these are used for (in terms of disease or conditions). There are, however several factors/areas where the information can be improved. My comments below relate to the highlighted text of the ms.
Amnat Charoen Healers in Thailand and their Medicinal Plants
Auemporn Junsongduang, Surapon Seansuk, Henrik Balslev
General remarks:
1. It is superfluous to include plant name authorities in text and appendix all times. Please say that authorities follow e.g. POWO (plantsoftheworldonline.org) and use only Latin names (with common Thai names) in the text and Appendix.
2. The common Thai name should be included in the text for the two or three most commonly used plants.
3. For most journals Introduction follows Materials and Methodology, Results, Discussion and conclusions
Section 4 should come after Conclusion
1. Line 89 – the highlighted sentence is not substantiated by a reference. Provide a reference or give an estimate of the medicinal plants used in Thailand, with refs)
2. As there is disparity between the healers give the age differences of the make and female healers here in Results
3. Table 1 – the males are from the same village. Their knowledge of the first (77 yrs) knew 130 med plants compared to the second (69 yrs) with knowledge of only 21 plants – is there an explanation to this? Please discuss under Discussion.
4. Line 107-Check spelling of Euphorbiaceae
5. It is unclear in this section about the exotic shrubs. Please include a separate paragraph on the exotic shrubs/herbs etc so that is is clear where these come from and where are these grown/imported
6. I am not familiar with the words bolus and many may not be generally; please just say what it is
7. Table 4. In your material and methods, please explain what is a decoction (cooked in water or infused in hot water?)
8. Table 4. Fresh – is this actually fresh or dried material – in Table 4, there is no category of dried material. Do you mean collected fresh or used fresh. Please explain.
9. Line 156. What family does Clinacanthus nutans belong to?
10. Line 164, 165. Is it possible to elaborate on infections and infestations and nutritional disorders?
11. Adding and using Iraq, India and Morocco here, you need to explain why these families are more prominently used. (For e.g. climatic limitations)
12. Line 272, 273. This is not a true statement and should be elaborated. All parts of trees and herbs can be dried and stored for use. And if the leaves of the herbs are used, the flowers should set seed for the continuation of their annual life cycle. Please elaborate and reword this sentence.
13. Line 264, 265-is it possible to elaborate on what is meant by decoction (as pointed out earlier) and metabolic diseases?
14. Line 290 – are these fresh or dried plants?
15. Line 296 – does external use mean topical?
16. Line 367 – is there anything on blood disorders?
17. Line 377 – Change to Maneenoon & al.
18. Line 395 – do home gardens grow only native plants? or also exotic plants? please explain.
19. Line 416 – What are the plants that may be purchased from distant sources? This aspect needs to be included in the abstract
Please see the highlighted areas in the ms. which refer to my comments.

Author Response
Comments 1. It is superfluous to include plant name authorities in text and appendix all times. Please say that authorities follow e.g. POWO (plantsoftheworldonline.org) and use only Latin names (with common Thai names) in the text and Appendix.
Response 1: Edited “In the text all of plants name authorities were deleted but still exist in AppendixS1”
Comments 2. The common Thai name should be included in the text for the two or three most commonly used plants.
Response 2: The common Thai name has been added followed by the scientific name in the text. (indicated in red font)
Comments 3. For most journals Introduction follows Materials and Methodology, Results, Discussion, and conclusions.
Response 3: We followed the template of Plants journal.
Comments 4. Line 89 – the highlighted sentence is not substantiated by a reference. Provide a reference or give an estimate of the medicinal plants used in Thailand, with refs)
Response 4: Line 89 – This section is a result that we have to report what we found in this study. We don’t think we need references in the results section.
Comments 5. As there is disparity between the healers give the age differences of the male and female healers here in Results?
Response 5: Answer in Line 91 -92 “The healers’ age was moderately correlated with the number of medicinal plants species they knew and used (r=0.39)”. And Line 93-94 “Noteworthy gender disparities were observed in the use of medicinal plant species among the healers, with males exhibiting statistically significant differences compared to females (p≤0.05) and they all engaged in agriculture.” However, we added more explanation Line 223-262.
Comments 6. Table 1 – the males are from the same village. Their knowledge of the first (77 yrs) knew 130 med plants compared to the second (69 yrs) with knowledge of only 21 plants – is there an explanation to this? Please discuss under Discussion.
Response 6: We added more explanation line 195-222. “The observed disparity in medicinal plant knowledge between the older healer (77 years) and the younger one (69 years) was moderately correlated with the number of medicinal plants species they knew and used and they can be attributed to several socio-cultural and environmental factors. First, generational knowledge transfer plays a significant role in traditional healing practices. The older healer likely grew up in a time when reliance on medicinal plants was more prevalent due to limited access to modern healthcare. This context may have necessitated a more extensive and experiential understanding of local biodiversity, which was integral to community health. Conversely, the younger healer’s reduced repertoire of plant knowledge may reflect a generational shift influenced by the increased availability of modern medicine, leading to a diminished reliance on traditional remedies [30,32]. Second, socio-economic changes could contribute to the knowledge gap. Urbanization, migration, and the younger generation's pursuit of formal education or alternative livelihoods often disrupt the intergenerational transmission of traditional ecological knowledge. Studies in similar contexts have demonstrated that elder healers act as critical knowledge repositories, but their ability to transfer this knowledge depends on sustained interaction with apprentices or younger community members [31].Third, traditional healers in Thailand can be categorized into six types: spirit healers (moh pao), midwives (moh tamyae), herbalists (moh samoonphrai), ritual specialists (moh sut khwan), traditional masseurs (moh nuad), and spiritual healers (moh tham). Some traditional healers possess diverse skills. Each healer has their own unique knowledge and methods for treating the same illness, which depends on the knowledge inherited from their ancestors and self-study. Most of this knowledge has been passed down through generations. The healing process primarily relies on local resources, beliefs, and rituals within the community. Finally, the loss of biodiversity due to environmental degradation in rural areas might also limit the younger healer’s opportunities to practice and expand their medicinal plant knowledge. Habitat destruction reduces the availability of certain plants, thereby narrowing the scope of traditional practices [28,31].”
Comments 7. Line 107-Check spelling of Euphorbiaceae
Response 7: Edited “Euphorbiaceae”
Comments 8. It is unclear in this section about the exotic shrubs. Please include a separate paragraph on the exotic shrubs/herbs etc so that is is clear where these come from and where are these grown/imported.
Response 8: We added more explanation line 477-495. “Among the medicinal plants used by Phu Tai healers, several exotic species were identified, representing 15% of the total species. These exotic plants, including shrubs, herbs, climbers, and other life forms, are predominantly grown in home gardens or sourced from community forests and, in rare cases, national parks. Examples include Andrographis paniculata (native to India, Nepal, and Sri Lanka), Aloe vera (originating from Northern Oman and Africa), and Catharanthus roseus (from Madagascar), which are cultivated in home gardens for their accessibility and ease of use. Exotic shrubs such as Barleria prionitis (native to Tropical Africa, Madagascar, and Asia and Punica granatum (originating from Türkiye to Northern Pakistan) are grown primarily in home gardens. They are valued for treating digestive system disorders, muscular pain, and other ailments. Herbs like Artemisia vulgaris (native to Eurasia, Indo-China, and North Africa) and Chromolaena odorata (native to Tropical and Subtropical America) are also common, with their parts harvested for treating sensory system disorders, injuries, and women's health conditions. The relocation of these species from their original habitats to home gardens is often due to their dwindling availability in the wild, resulting from overharvesting and habitat destruction. In some cases, exotic plants such as Cestrum nocturnum (native to Mexico and Venezuela) are purchased from distant areas to meet local medicinal needs, as they have become scarce in community forests. The introduction of these plants reflects both their perceived medicinal value and the adaptability of traditional healers in integrating foreign species into their practices. The Original data or native rank of exotic medicinla plants in this study from POWO [https://powo.science.kew.org/].”
Comments 9. I am not familiar with the words bolus and many may not be generally; please just say what it is?
Response 9: a bolus (a soft, compact mass used for medicinal purposes) is one traditional preparation method often used for ease of administration. Here's how it is typically done: Method for Preparing a Bolus Ingredients Selection: Collect the required medicinal plants based on the ailment to be treated. Use specific plant parts (roots, leaves, bark, seeds, etc.) depending on the traditional knowledge or recipe. Processing: Wash the plant materials thoroughly to remove dirt and contaminants. Dry the materials if needed (e.g., sun-drying or shade-drying to retain active compounds). Grind the plant materials into a fine powder using a mortar and pestle or a mechanical grinder. Mixing: Combine the powdered plant material with a binder (e.g., honey, water, or a starch paste) to form a malleable mass. Sometimes additional ingredients like spices (e.g., turmeric) or sugars are added to enhance taste or efficacy. Shaping: Shape the mixture into small, rounded balls (boluses) of uniform size for easy ingestion. Traditionally, the boluses are made small enough to swallow without difficulty. Storage: Store the boluses in a cool, dry place, often wrapped in leaves or stored in airtight containers. In some traditions, boluses are coated with powdered herbs to prevent sticking or enhance preservation. Applications of Boluses: They are typically used for conditions requiring long-term medication, as boluses allow for controlled dosing. Common ailments treated with boluses include digestive issues, fever, or chronic inflammatory conditions.
Comments 10. Table 4. In your material and methods, please explain what is a decoction (cooked in water or infused in hot water?).
Response 10: Decoction means the way the healers boiled medicinal plants in hot water and drink like a tea. A decoction is a method of extracting active compounds from medicinal plants by boiling them in water. This preparation technique is commonly used in traditional medicine to make herbal remedies.
Comments 11. Table 4. Fresh – is this actually fresh or dried material – in Table 4, there is no category of dried material. Do you mean collected fresh or used fresh. Please explain.
Response 11: Some medicinal plants the healers used to curing by eat like a vegetable as fresh (collected and used fresh).
Comments 12. Line 156. What family does Clinacanthus nutans belong to?
Response 12: Added (Acanthaceae) line 158.
Comments 13. Line 164, 165. Is it possible to elaborate on infections and infestations and nutritional disorders?
Response 13: Since we follow Cook (1995) we prefer not to diverge from that standard to avoid confusion, so we prefer not to elboartae on infections and. Infestations and nutritional disorders.
Comments 14. Adding and using Iraq, India and Morocco here, you need to explain why these families are more prominently used. (For e.g. climatic limitations).
Response 14: Added more explanation line 274-298. “Fabaceae's global dominance, as seen in Thailand, Namibia, and parts of India, is attributed to its ecological versatility, nitrogen-fixing ability (allowing it to grow in poor soils), and rich secondary metabolites. Its species often exhibit wide pharmacological activity, making it a universally relied-upon family for traditional medicine. Similarly, Zingiberaceae’s aromatic rhizomes have proven effective across tropical regions for digestive and anti-inflammatory uses, reflecting their adaptation to these climates and associated diets. However, in Iraq Lamiaceae was the family with the most medicinal plant species [23]. It is likely related to its adaptability to arid and semi-arid climates, which characterize much of the region. Lamiaceae species, such as mint and thyme, thrive in these conditions and are widely recognized for their essential oils with antimicrobial and anti-inflammatory properties (Jamshidi-Kia et al., 2018). In India, the Euphorbiaceae family, prominent in Kerala [21]. and Tamil Nadu [68], is ecologically abundant in tropical and subtropical regions. The family includes species like Phyllanthus and Euphorbia, which are known for their pharmacological properties, including anti-diabetic, hepatoprotective, and anti-inflammatory effects (Joseph & Raj, 2011). The diverse uses of Euphorbiaceae in India also reflect its cultural integration into Ayurvedic medicine, where its species are used for treating chronic diseases. In Morocco, the prominence of Asparagaceae, specifically for dermatological conditions [69], is likely tied to the region’s Mediterranean climate. Species such as Asparagus officinalis and Dracaena thrive in this environment and are rich in saponins and antioxidants, which are effective in skin-related therapies (Belkhir et al., 2021). The prominence of certain plant families in traditional medicinal systems is influenced by a combination of climatic conditions, ecological availability, cultural practices, and the specific therapeutic needs of local populations. This demonstrates how environmental conditions and prevalent health issues shape the medicinal plant preferences of local populations.
Comments 15. Line 272, 273. This is not a true statement and should be elaborated. All parts of trees and herbs can be dried and stored for use. And if the leaves of the herbs are used, the flowers should set seed for the continuation of their annual life cycle. Please elaborate and reword this sentence?
Response 15: We rewrote lines 323-328. “A total of nine different life forms of medicinal plants were identified among the 15 healers. The most commonly used life form was shrubs, which is consistent with a study conducted in southern Thailand [24]. However, this finding contrasts with another study in northeastern Thailand, specifically in Nakhon Phanom and Roi Et provinces, where trees were the most prevalent life form used for medicinal purposes [31]. The advantage of trees lies in their multiple usable parts, which can be harvested repeatedly, allowing for sustainable collection. In contrast, herbs, which are used in their entirety, are typically consumed once and may be at risk of overharvesting, potentially leading to extinction [31]. Following shrubs, herbs—including both native and exotic species—and herbaceous climbers were commonly utilized. Additionally, two species of palms, including a climbing palm, were recognized for their medicinal properties. Unique to this study, one species each from the life forms bamboo, exotic aquatic herb, and epiphytic fern were also documented as having medicinal uses. For comparative purposes, studies in Guangxi Fangcheng, China, and Tamil Nadu, India, found that herbs were the predominant life form used in traditional medicine [66, 68], highlighting regional differences in plant selection based on ecological and cultural factors.”
Comments 16. Line 264, 265-is it possible to elaborate on what is meant by decoction (as pointed out earlier) and metabolic diseases?
Response 16: Decoction is a method for preparation of medicinal plants for curing but metabolic diseases is a term of Health Disorders that we followed by Cook (1995).
Comments 17. Line 290 – are these fresh or dried plants?
Response 17: Soaking the plants in the water for curing the patient can used both fresh or dried part of medicinal plants.
Comments 18. Line 296 – does external use mean topical?
Response 18: No, it is mean healers used medicinal plant externally, such as applied to smear to the skin, compress on the skin or wound.
Comments 19. Line 367 – is there anything on blood disorders?\
Response 19: blood disorders including Anaemia and Coagulation (Inta et al., 2008)
Comments 20. Line 377 – Change to Maneenoon & al.
Response 20: Edited “Maneenoon et al.” line 438.
Comments 21. Line 395 – do home gardens grow only native plants? or also exotic plants? please explain.
Response 21: Both native and exotic plants were grown in home garden. We provide more explanation in line 477-495. “Among the medicinal plants used by Phu Tai healers, several exotic species were identified, representing 15% of the total species. Especially exotic plants, including shrubs, herbs, climbers, and other life forms, are predominantly grown in home gardens or sourced from community forests and, in rare cases, national parks.”
Comments 22. Line 416 – What are the plants that may be purchased from distant sources? This aspect needs to be included in the abstract. Please see the highlighted areas in the ms. which refer to my comments.
Response 22: We added in abstract line 22-23. Medicinal plants that are purchased 1. Hellenia speciosa 2. Cinnamomum parthenoxylo 3. Talinum paniculatum
Round 2
Reviewer 3 Report
Comments and Suggestions for Authors
Authors: The manuscript is greatly improved in its revised form, and is publishable as a useful and informative paper on SE Asian medical plants.
Please add in your final submission in text and Table: Plant names (not plants) according to POWO [plantsoftheworldonline.org]
Author Response
Comment 1: Please add in your final submission in text and Table: Plant names (not plants) according to POWO [plantsoftheworldonline.org]
Reply 1: All plant names in the manuscript follow WorldFloraOnline